# Medieval Monastery Gardens in Iceland and Norway

Per Arvid Åsen 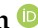

Natural History Museum and Botanical Garden, University of Agder, 4604 Kristiansand, Norway;
per.arvid.aasen@uia.no

**Abstract:** Gardening was an important part of the daily duties within several of the religious orders in Europe during the Middle Ages. The rule of Saint Benedict specified that the monastery should, if possible, contain a garden within itself, and before and above all things, special care should be taken of the sick, so that they may be served in very deed, as Christ himself. The cultivation of medicinal and utility plants was important to meet the material needs of the monastic institutions, but no physical garden has yet been found and excavated in either Scandinavia or Iceland. The Cistercians were particularly well known for being pioneer gardeners, but other orders like the Benedictines and Augustinians also practised gardening. The monasteries and nunneries operating in Iceland during medieval times are assumed to have belonged to either the Augustinian or the Benedictine orders. In Norway, some of the orders were the Dominicans, Fransiscans, Premonstratensians and Knights Hospitallers. Based on botanical investigations at all the Icelandic and Norwegian monastery sites, it is concluded that many of the plants found may have a medieval past as medicinal and utility plants and, with all the evidence combined, they were most probably cultivated in monastery gardens.

**Keywords:** medieval gardening; horticulture; monastery garden; herb; relict plants; medicinal plants

## 1. Introduction

Monasticism originated in Egypt's desert, and the earliest monastic gardens were vegetable gardens (McLean 1989; Meyvaert 1986). In approximately 350–400AD, organized monasticism spread from the Eastern Mediterranean to Italy, France and Spain and North Africa (Schumacher 2009). The rule of St. Benedict (c. 480–550) stated that the monastery should, if possible, be so arranged that all necessary things, such as water, mill, garden, and various crafts may be situated within the enclosure, so that the monks may not be compelled to wander outside, for that is not at all expedient for their souls. It is worth noting that the Benedict's rule was translated into the Old Norse language (Myking 2017). Further, the monastic rule of St. Isidore, bishop of Seville (c. 560–636), specified the need for a garden in the cloister (Harvey 1990).

Several thousand monasteries were founded in Europe, the northernmost located in Greenland (Grayburn 2015), Iceland (Kristjánsdóttir 2017) and in central Norway (Lunde 1987).

The plan of St. Gall (www.stgallplan.org, accessed on 12 January 2021) is the oldest surviving plan of a complete monastery. It originated in the Benedictine monastery Reichenau c. 819–826, and is now kept in St. Gallen monastery in Switzerland. The plan is probably a design for an ideal monastic community, complete with a herb garden by the infirmary, a kitchen garden and an orchard associated with the cemetery; for details, see Tremp (2014). Another plan is that of the Priory of Christ Church at Canterbury c. 1165, with the herb garden located near the infirmary, enclosed between wattled fences, and a tree garden included in the cemetery (Harvey 1990). Both plans show the cloister garden at the heart of the monastery, alongside the church. It was dominated by a green lawn, maybe with violets, lillies, strawberries and dasies symbolizing the virgin Mary (Behling 1967; Widauer 2009), an evergreen tree, a rose and a fountain. It was a place of retreat, and where the monks processed at regular intervals (Landsberg 1998; Stannard 1983).

The herb garden included general medicinal plants, those that are poisonous, narcotics, plants related to blood-letting needs, aromatics and some ornamentals that would refresh patients by their beauty (Landsberg 1998). The inventory would obviously vary with the different monastic communities. Not least would this apply to the Icelandic and Norwegian monasteries located at the northern borders of Christian civilization, where the establishments were smaller and the climate less favourable.

However, before the monasteries were established, there existed viking gardens, gardens in towns and possibly also at medieval farmsteads (e.g., Helweg 2020; Holmboe 1921; Sandvik 2006; Sjögren et al. 2021; Øye 2015). It seems obvious that the religious orders in Iceland and Norway would utilize this knowledge when monastic gardens were established. Both Norway and Iceland were highly influenced by Anglo-Saxon and Irish Christianity, where monastic gardens were common (Coppack 2006; Larsson et al. 2012).

In this context, a garden is understood as a fenced-in area where plants were cultivated at a small scale using only handheld tools (spade)—this in contrast to agriculture, with large crops in wider fields, where the plough was the main tool.

In Iceland, Kristjánsdóttir (2017) has described fourteen monasteries and nunneries, dated between 1030 and 1554, all assumed to have belonged to either the Augustinian or the Benedictine orders (Figure 1).

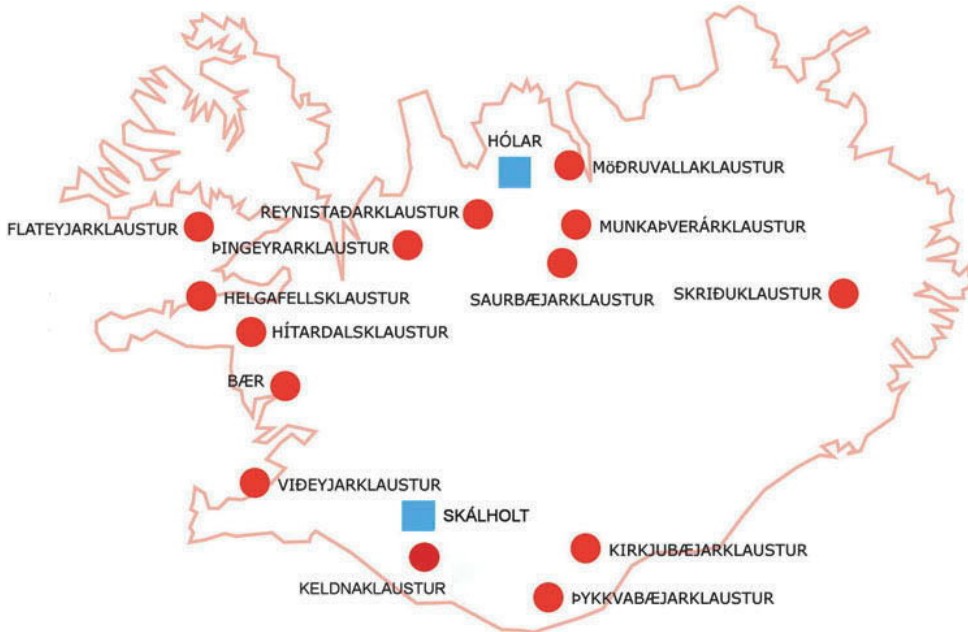

**Figure 1.** Medieval Icelandic monasteries and nunneries (red), and bishoprics (blue).

In the first complete description of the 26 Norwegian monasteries and five nunneries, Lange (1847) stated that «*All the monasteries had a garden, often several, and they were well tended...the monks brought fruit-trees, cuttings, herbs and flowers from abroad, in order to plant them in Norwegian soil. And still today one can find gardens by the monasteries that contain fruit-trees*». The monasteries were all dissolved during the Lutheran Reformation c. 1536.

Archaeobotanical material from excavations in the medieval towns in Norway (e.g., Buckland and Wallin 2017; Dunlop and Sandvik 2004; Eriksson 1990; Griffin 1977, 1981, 1988; Hjelle 2007; Lindh et al. 1984; Moltsen 2016a, 2016b; Petersén and Sandvik 2006; Sandvik 2000a, 2000b; Sture 2017a, 2017b, 2017c; Wallin 2017) and six monastic sites in Iceland (https://notendur.hi.is/~sjk/KK.htm, accessed on 10 February 2021) give important evidence on the kind of plants that were present in the Middle Ages, and possible cultivation in monastic gardens (Figure 2).

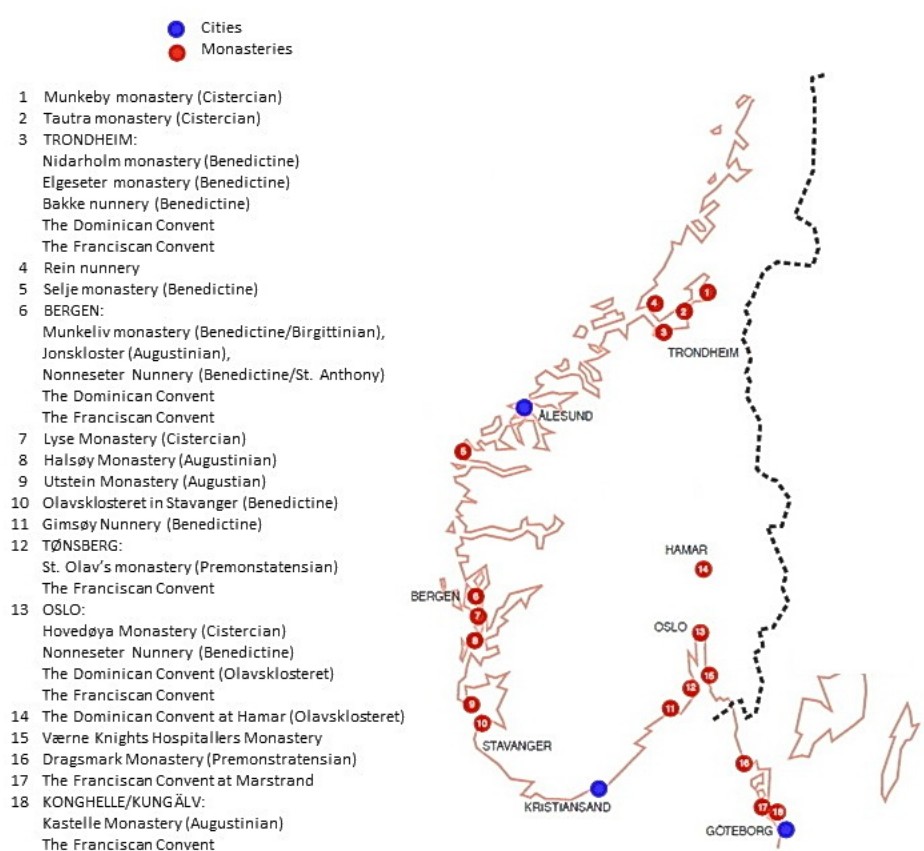

**Figure 2.** Medieval Norwegian monasteries, nunneries and convents.

Medieval cultural relict plants are considered living remnants from medieval times, i.e., they have survived at a certain locality since medieval times. Løjtnant (2017) has investigated the relict flora of 2600 medieval localities in Denmark between 1993 and 2011. He has shown that a particular medieval relict flora exists that, in his opinion, dates back at least to medieval times. Problems always exist in interpreting which species can be termed a medieval relict plant. I have focused on introduced plants, the isolation of a given locality, the presence of archaebotanical records, finds in other similar loacalities and literature records.

This article is based on botanical surveys of all the Icelandic and Norwegian monastic sites between 2003 and 2014 and known archaeobotanical data (Larsson et al. 2012; Kristjánsdóttir et al. 2014; Åsen 2015) in order to increase the knowledge of the Icelandic and Norwegian monastic gardens and plants to be able to compare and interpret these results with prevailing opinions about the monastic garden, cultural relict plants, and the plant material found in similar contexts in other parts of northern Europe.

The main purpose of this research—*from a strictly botanical view*—is to to establish a possible connection between introduced medieval utility plants, relict plants, archaeobotanical data and written sources with possible monastery gardens in Iceland and Norway, suggesting foremost utility gardens. The scope of the research does not focus on garden design or the aestetics of gardens, since no monastery garden has been exactly located or excavated and very little archaeobotanical data exists from the actual monastic sites.

## 2. Results

### 2.1. Iceland

**The Benedictine Monastery at Bær í Borgarfirði** was established by the English missionary bishop Rúðólfur in 1130 as the first monastery in Iceland (Bæjarklaustur). Three monks were still present in 1149, when Rúðólfur became abbot of the Abington monastery in England and the monastery at Bær ceased. It is suggested the monastery was probably located beneath the present church (Kristjánsdóttir 2014).

Not far from the church, *Laukaflatir*, a plain approximately 100 × 100 m, is completely dominated with field garlic (*Allium oleraceum*), an introduced species known only from Bær and Skáney today, and the first record from Bær is in Halldórsson (1783). The place name Laukaflatir has been known for a very long time (Davíðsson 1943). There is an unverified story that Rúðólfur brought the field garlic to Bær from the Trondheim area in Norway. (At the the Tautra monastery on an island in Trondheimsfjord, the field garlic is common.) At Skáney, a German doctor, Lazarus Mattheusson, practised in the period 1527–1570. He could have introduced the field garlic from Germany or maybe from Bær (Larsson et al. 2012). Table 1 provides a list of the utility plants mentioned in the text.

**Table 1.** Utility plants mentioned in the text possibly cultivated in Icelandic and Norwegian medieval monastery gardens.

| Scientific Name | English Name | Norwegian Name | Icelandic Name |
|---|---|---|---|
| *Aegopodium podagraria* | ground-elder | skvallerkål | geitakál |
| *Aethusa cynapium* | fool's parsley | hundepersille | villisteinselja |
| *Allium oleraceum* | field garlic | vill-løk | villilaukur |
| *Allium scorodoprasum* | sand leek | bendelløk | |
| *Allium ursinum* | ramsons | ramsløk | bjarnarlaukur |
| *Anchusa officinalis* | alkanet | oksetunge | nautatunga |
| *Angelica archangelica* ssp. *archangelica* | garden angelica | fjellkvann | ætihvönn |
| *Aquilegia vulgaris* | columbine | akeleie | skógarvatnsberi |
| *Arctium lappa* | greater burdock | storborre | krókalappa |
| *Arctium minus* | lesser burdock | småborre | |
| *Artemisia absinthium* | wormwood | malurt | malurt |
| *Artemisia vulgaris* | mugwort | burot | búrót |
| *Asparagus officinalis* | asparagus | asparges | spergill |
| *Asperugo procumbens* | madwort | gåsefot | gæsalöpp |
| *Ballota nigra* | black horehound | hunderot | |
| *Balsamita major* | costmary | balsam | maríubrá |
| *Bellis perennis* | daisy | tusenfryd | fagurfífill |
| *Berberis vulgaris* | barberry | berberis | ryðbroddur |
| *Borago officinalis* | borage | agurkurt | |
| *Brassica oleracea* | cabbage | kål | garðakál |
| *Campanula rapunculoides* | creeping bellflower | ugrasklokke | skriðklukka |
| *Cannabis sativa* | hemp | hamp | hampjurt |
| *Carum carvi* | caraway | karve | kúmen |
| *Chelidonium majus* | greater celandine | svaleurt | svölujurt |
| *Cichorium intybus* | chickory | sikori | sikoría |
| *Conium maculatum* | giftkjeks | hemlock | eitruð plöntutegund |
| *Corylus avellana* | hazel | hassel | hesli |
| *Crataegus monogyna* | hawthorn | hagtorn | snæþyrnir |
| *Cynoglossum officinale* | hound's tongue | hundetunge | hundatunga |
| *Daucus carota* | wild carrot | villgulrot | gulrót |
| *Descurainia sophia* | flixweed | hundesennep | þefjurt |
| *Digitalis purpurea* | foxglove | revebjelle | fingurbjargarblóm |
| *Euonymus europaeus* | spindle | spolebusk | beinviður |
| *Fagopyrum esculentum* | buckwheat | bokhvete | bókhveiti |
| *Fagus sylvatica* | beech | bøk | beyki |
| *Foeniculum vulgare* | fennel | fennikel | fennika |
| *Fraxinus excelsior* | ash | ask | evrópuaskur |
| *Fumaria officinalis* | common fumitory | jordrøyk | reykjurt |
| *Glechoma hederacea* | ground-ivy | korsknapp | krosshnappur |
| *Hedera helix* | common ivy | bergflette | bergflétta |
| *Hesperis matronalis* | dame's violet | dagfiol | næturfjóla |
| *Hyoscyamus niger* | henbane | bulmeurt | |
| *Hyssopus officinalis* | hyssop | isop | ísópur |
| *Lamium album* | white dead-nettle | dauvnesle | ljósatvítönn |
| *Leonurus cardiaca* | motherwort | løvehale | |
| *Linum usitatissimum* | flax | lin | spunalín |
| *Lithospermum officinale* | common gromwell | legesteinfrø | |

**Table 1.** *Cont.*

| Scientific Name | English Name | Norwegian Name | Icelandic Name |
|---|---|---|---|
| *Malus domestica* | apple | hageeple | epli |
| *Malus sylvestris* | crab apple | villeple | villiepli |
| *Malva neglecta* | dwarf mallow | småkattost | |
| *Marrubium vulgare* | white horehound | borremynte | vallarhélukrans |
| *Myrica gale* | bog myrtle | pors | mjaðarlyng |
| *Nepeta cataria* | cat-mint | kattemynte | kattablom |
| *Papaver somniferum* | opium poppy | opiumvalmue | draumsól |
| *Pastinaca sativa* | parsnip | pastinakk | pastínakka |
| *Petasites hybridus* | butterbur | legepestrot | hestafífill |
| *Peucedanum ostruthium* | masterwort | mesterrot | |
| *Pisum sativum* | garden pea | ert | gráerta |
| *Plantago major* | greater plantain | groblad | græðisúra |
| *Primula veris* | cowslip | marianøkleblom | sifjarlykill |
| *Prunus cerasus* | dwarf cherry | surkirsebær | súr kirsuber |
| *Prunus padus* | bird cherry | hegg | heggur |
| *Pyrus communis* | pear | pære | perur |
| *Ribes rubrum* | red currant | hagerips | rauðberjarifs |
| *Ribes uva-crispa* | gooseberry | stikkelsbær | stikilsberjarunni |
| *Rosa rubiginosa* | sweet-briar | eplerose | |
| *Sambucus nigra* | elder | svarthyll | svartyllir |
| *Sanguisorba officinalis* | greater burnet | blodtopp | blóðkollur |
| *Saponaria officinalis* | soapwort | såpeurt | þvottajurt |
| *Solanum dulcamara* | bittersweet | slyngsøtvier | náttskuggi |
| *Solanum nigrum* | black nightshade | svartsøtvier | húmskuggi |
| *Tanacetum vulgare* | tansy | reinfann | regnfang |
| *Urtica dioica* | common nettle | stornesle | brtenninetla |
| *Urtica urens* | small nettle | smånesle | smánetla |
| *Valeriana officinalis* | valerian | legevendelrot | garðabruða |
| *Verbascum densiflorum* | dense-flowered mullein | prydkongslys | |
| *Verbascum nigrum* | dark mullein | mørkkongslys | surtarkyndill |
| *Verbascun thapsus* | great mullein | filtkongslys | gullkyndil |
| *Veronica beccabunga* | brooklime | bekkeveronika | veitarbládepla |
| *Vica faber* | broad bean | hestebønne | hestabaunir |
| *Viola odorata* | sweet violet | marsfiol | ilmfjóla |

Most likely *laukr* (leek, onion, garlic) was cultivated in Iceland during the Middle Ages. Guðrún Ósvífrsdóttir converses with her sons in the laukagarðr (leek garden) near the Helgafell monastery. Further, a laukagarðr is mentioned at Holar, known between 1457 and 1525, where the bishop is said to have died in 1457. The only pollen record, from Skriðuklaustur (see below), indicates cultivation in Iceland (Jensson 2004), although pollen from a single place does not prove conclusively that it was grown in Iceland at the time. Laukagarðr is also mentioned in Jónsbok, but this may originate in Norwegian laws (Larsson et al. 2012).

In a recent pollen study from Bær, *Allium* was not recorded. Pollen of *Brassicaceae*, *Hordeum*-type (barley) and *Polygonum aviculare* may indicate cultivation. However, according to Riddell and Erlendsson (2015a), the only definite conclusion from the rapid scanning of pollen is that cereal grain was present at Bær at some point in the past.

**Þingeyraklaustur í Húnaþingi (1133–1551).** The present church, farm and Benedictine monastery site are situated upon the summit of a ridge. Today, the primary land use is grazing for horses and hay fields. The flora surrounding the church is dominated by rather common plant species, except for madwort (*Asperugo procumbens*), found growing on open, stirred soil east of the church in 2009 and 2010. The madwort, an annual, first recorded from Þingeyrar in 1929, may sprout from dormant seeds in the soil when stirred. It is suggested

as a relict utility plant of medieval monastic cultivation in Norway (Åsen 2015); in Denmark, madwort is listed as a good indicator of medieval gardening (Løjtnant 2017).

The pollen record for Þingeyrar shows that from foundation, the monastery was active in altering the character of its immediate environs to a pastoral landscape, probably through livestock grazing, perhaps by deliberate scrub clearance, but also through other forms of resource use, e.g., dwarf birch (*Betula nana*) for fuel (Riddell et al. 2018), and possible development of a monastery garden.

**Munkaþverárklaustur í Eyjafirði.** The Benedictine monastery (1555–1551) was arranged in a traditional manner, with the church forming the northern range of the cloister garth. It was located near the present church and in front of the farm (Kristjánsdóttir 2017).

No monastery garden is known. However, it is interesting to refer to the monastery's first abbot, Nikulás Bergsson (1155–1159), who made a pilgrimage to Jerusalem in approximately the period 1149–1153 (Jensen 2004). In his itinerary, he mentions the good doctors of Salerno. This could indicate that he had some knowledge of medicine and medicinal herbs, and it is tempting to suggest a possible herb garden once placed along the productive banks of Þverá.

An inventory of the monastery, dated 1525, listed both a mustard seed and pepper seed grinder (Kristjánsdóttir 2017). In an Icelandic cookbook, dated from the last quarter of the 15th century, recipes with both mustard and pepper are presented. Grinding mustard seeds with honey and vinegar made a good dressing that could keep for forty days (Grewe and Hieatt 2001; Veirup 1993). Mustard was an important ingredient in the medieval kitchen, both as medicine and condiment, and the seeds could have been imported or harvested in the monastery herb garden. Both white and charlock mustard (*Sinapis alba* and *arvensis*) are known in Iceland (Wąsowicz et al. 2013).

**Hítardalsklaustur á Mýrum (Benedictine 1166–1201/1237).** Georadar investigations in 2013 indicated possible traces of ruins in the ground north of the farm. Further, a cornerstone with a carved face exists above ground (Kristjánsdóttir 2014). Meadows and grazing fields dominate the area around the farm, with common species and commonplace courtyard vegetation.

**Þykkvabæjarklaustur í Álftaveri (Augustinian 1168–1548).** The monastery area, surrounded by hayfields, is located like a green island on Mýrdalssandur southeast of Mýrdalsjökull. Georadar investigations in 2015 suggested a traditional cloister garth buried underneath Klausturhóll and Fornufjós, north of the present church (Kristjánsdóttir 2017).

The Augustians are well known for hospitals and medicinal herb gardens, e.g., Æbelholt in Denmark (Møller-Christensen 1982) and Soutra Aisle in Scotland (Moffat 1995). We may speculate that the present findings at Þykkvabæjarklaustur, e.g., the sheer size of the ruins could indicate the presence of a monastic herb garden.

Our records of the flora in the area present common species, including the archaeophyte caraway (*Carum carvi*; Larsson et al. 2012). The use of caraway in Europe dates back at least to Roman times. In addition, it was found at four other monastic sites in Iceland—Flateyarklaustur, Kirkjubæjarklaustur, Möðruvallaklaustur and Viðeyjarklaustur—growing in courtyards and pastures. Pollen finds at Mývatn from the period 1000–1300 may suggest medieval introduction (Kristjánsdóttir et al. 2014).

**Flateyjarklaustur á Breiðafirði.** The Augustinian monastery at Flatey (1172–1184) was probably located northeast of the present church, at Klausturhólar.

At Flatey, greater plantain (*Plantago major*), Icelandic græðisúra, was recorded. It was an important medicinal plant in Scandinavia, and well known in the Middle Ages, but it is rare in Iceland today. It is probably a cultural relict plant. Its vernacular name implies that its value as a medicinal plant was known before written evidence about the plant's virtues and properties reached us. Pollen records are known from several sites, including Skriðuklaustur and Viðey; in addition, plants were observed at Viðey (Larsson et al. 2012). Further, caraway was found.

**Helgafellsklaustur í Helgafellssveit** was the Augustinian monastery moved from Flatey in 1184 and dissolved 1543. Located south of Stykkishólmur, the buildings probably

stood beneath the present church and cemetery, with no traces above ground today. The monastery was rich with extensive land holdings, livestock, objects and books, hosted a school, produced beer, books, iron and textiles, and provided housing for lay people (Kristjánsdóttir 2017). In the Saga of Laxdæla, a leek garden is mentioned near the monastery.

Today, the area around the church and farm is dominated by meadows (*graslendi*) with commonplace vegetation. A pollen sample from a possible floor layer within a medieval building presents a matrix of habitat types: grassland, heathland and woodland, with wetland perhaps the most dominant. The pollen analysis revealed little evidence of plants with the potential for medicinal application or utility, except for a single grain of *Artemisia*-type pollen (Riddell and Erlendsson 2015b).

The species of *Artemisa* are well-known medicinal herbs, even back in Antique times. Several other archaeobotanical finds, including written accounts that report mugwort (*Artemisia vulgaris*) growing in Iceland in the 17th century, and cultivation near Skálholt in the 18th century, may indicate possible cultivation and use as a medicinal herb (Larsson et al. 2012; Riddell and Erlendsson 2015b).

**Kirkjubæjarklaustur á Síðu.** The Benedictine nunnery at Kirkjubær (1186–1542) has been partly excavated, but no specific monastic buildings have been revealed. The archaeophyte caraway was common at the site.

**Saurbæjarklaustur í Eyjafirði (Augustinian 1200–1224).** The area around farm and church was dominated by commonplace vegetation, except for flixweed (*Descurainia sophia*), found among a heap of rocks south of the present church. The rocks could be either parts of old ruins or placed there in order to level the surrounding fields (Larsson et al. 2012; Kristjánsdóttir 2017). Although flixweed has probably been used as a medicinal herb and has been recorded with many archaeobotanical finds in Europe, including Scandinavia, it is difficult to have an opinion concerning its present status in Iceland. No archeobotanical record exists, so conclusions are rather tentative. It is characterized as a neophyte by Wąsowicz et al. (2013), with the first Icelandic record in 1889.

The Augustinian **Viðeyjarklaustur**, 1226–1539, was one of the wealthiest monasteries in Iceland. Excavations of a farm mound took place in 1987–1995, including archaeobotanical investigations (Bjarnardóttir 1997; Hallgrímsdóttir 1993; Hallsdóttir 1993). Introduced utility plants found that could have been cultivated were *Artemisia*, *Myrica gale*, *Plantago major*, *Sanguisorba officinalis*, *Urtica dioica* and *Valeriana officinalis*. Later investigations have shown that the Viðey monastery probably stood on the northwestern part of the farm mound (Kristjánsdóttir 2017).

Further pollen analysis was performed by Riddell and Erlendsson (2014) in order to look for possible utility plants. They noted *Plantago major* and *Valeriana officinalis* as possible evidence of a monastic garden. In addition, we observed greater plantain as well as a large population of caraway (Larsson et al. 2012).

A small statue of St. Dorothy—the patroness of gardeners—was found during the excavations. She is regarded as the patroness of gardeners, known both in Iceland and Norway (Wolf 1997). This finding may be an indication that gardening was part of the work at the monastery (Kristjánsdóttir et al. 2014). A similar statue was found during the recent excavations in medieval Oslo (see below).

**Reynistaðarklaustur í Skagafirði** was a Benedictine nunnery (1295–1551), with ruins located underneath the farm mound southwest of the present church. Pollen analysis revealed that the presence of *Hordeum*-type pollen along with the occurrence of other plant species and taxa associated with disturbed ground might suggest that cereal cultivation was a feature of the land management regime in the past (Riddell and Erlendsson 2015c). Further, pollen of the *Brassicaceae* family was recorded. However, they concluded that there was no definitive evidence of other plant species that harbor utilitarian properties in the pollen assemblage.

**Möðruvallaklaustur í Hörgárdal.** The Augustinian monastery (1296–1551) burned in 1316 but was rebuilt in 1326, with the help of brethren from Elgeseter sister monastery in

Trondheim. The monastery probably stood beneath the present church and cemetery, but the buildings have not yet been found with certainty (Kristjánsdóttir 2017).

Further, at Möðruvallaklaustur, a mustard seed grinder was included in an inventory (together with pepper and malt grinder). Mustard seeds were either imported or perhaps harvested from *Sinapis* plants in a monastery garden (see Munkaþverárklaustur above). No archaeobotanical analysis has been published so far, and the only possible utility relict plant found was caraway.

**Skriðuklaustur í Fljótsdal (1493–1554).** The excavations of the Augustinian monastery revealed a complete layout of a monastic building, consisting of several small cells, a church, and a cloister garth with a well, laid out in a manner similar to most medieval monastic buildings outside Iceland (Kristjánsdóttir 2013).

Archaeobotanical analyses have been summarized by Larsson et al. (2012). Of some twenty taxa, *Allium*, *Borago officinalis*, various species of the *Brassicaceae* family, *Plantago major*, *Urtica dioica* and *Urtica urens* could have been introduced as medicinal herbs, possibly cultivated in a monastery garden. Today, all have a restricted distribution in association with human activity. In addition, Shaw (2012) published a find of a charred seed tentatively identified as crab apple (*Malus sylvestris*), suggesting import for food. (Dried crab apple is eatable.) However, even if the climate was becoming cooler, cultivation could have been a possibility before the onset of the Little Ice Age c. 1550. This is somewhat in line with the Norwegian priest Ivar Bårdsson, who reported good-tasting apples growing in Greenland c. 1360. (Jonsson 1930).

The pollen analyses revealed a grass-dominated vegetation both before and after the monastic period, the herbs only present in between (Jensson 2004).

These herbs, together with the skeletons and surgical equipment found at the site, indicate strongly that the monastery may have functioned as a hospital (Kristjánsdóttir 2013). Following European tradition, the herb garden was probably a fenced-in area near the infirmary, outside the cloister garth. However, the excavations completed in 2011 gave no indication of its placement. Today, the area around the monastery ruins is dominated by meadows and grazing fields with commonplace vegetation.

### 2.2. Norway (Bohuslän Included until 1658)

**Tautra Cistercian monastery** was founded in 1207 on Tautra island, NE of Trondheim, as daughter of Lyse monastery. Only parts of the church are still standing. An inventory dated 1532 proves that the monastery once had extensive activities in agriculture and gardening, including the entire island (Ekroll 1996).

The apple garden was extant and described in 1613 (Nøvik 1901). In 1743, some gardens at Tautra included cherries, hazelnuts, hips and ash trees; and in the monastery ruins, there was a profusion of rare herbs (Nøvik 1901). Remnants of a large monastery garden south of the ruins, that included a grove of old apple trees, were described in 1774. Further, trees including ash, oak and hawthorn probably originated from the monastery garden. Several cherry and plum trees, pear, and garden berries, that were cultivated at the island, could be a possible continuation from monastic gardening (Schøning 1910). In 1807, cherries were still present in large quantities at Tautra (Nøvik 1901); and in 1808, the island was described as a place where the monastic garden herbs grew wild (Suhm 1808). In 1817, parts of the monastery garden could still be seen; and in 1879, the cloister garth was excavated (Ekroll 1996). At the start of the 20th century, nothing was left of the monastery garden (Nøvik 1901).

The cherries mentioned are probably all sour cherries (*Prunus cerasus*), and were most likely a monastic introduction, like the cherries introduced by the mother monastery at Lyse and growing wild at Utstein monastery in 1758. Crab apples (*Malus sylvestris*) and hawthorns (*Crataegus*) growing at the beach south of the ruins might also originate from the monastery, as well as the ashes and columbine (*Aquilegia vulgaris*) still common in the ruins. Other possible relict plants growing on Tautra today include *Allium oleraceum* (see Bær above), *Arctium minus*, *Berberis vulgaris*, *Carum carvi*, *Hesperis matronalis*, *Verbascum*

*densiflorum*, *Verbascum thapsus* and *Veronica beccabunga*. Some earlier finds may include *Asperugo procumbens*, *Balsamita major*, *Bellis perennis*, *Hyoscyamus niger*, *Papaver somniferum* and *Sambucus nigra*.

Today, **Reinskloster nunnery (c. 1226)** is part of Rein manor NW of Trondheim. Manorial gardens were established at least from 1762. An old tree garden, Gammelhagen, is located just south of the manor and ruins. Local history—at least since 1703—says that the nunnery garden originally included 24 ash trees (*Fraxinus excelsior*), brought by German nuns in the 13th century. Today, a few large ashes are still standing, probably third-generation trees, approximately 200 years old. However, a big tree that was cut down approximately fifty years ago had an age of approximately 700 years (Sundfør 1996). Since this part of the manorial property has been left undeveloped, it is suggested as an originally medieval garden relict.

**Nidarholm Benedictine Monastery (c. 1100)** is located at Munkholmen, just north of Trondheim harbor. The island was probably used for grazing after the dissolution until the 17th century, when the military acquired the property. Nothing is known of the presence of a monastery garden, but the area north and west of the cloister garth may have been used for cultivation. Further, some gardening may have taken place in the graveyard. Sour cherries are prominent on the island, a possible monastic garden relict. Further, a previous find of *Asperugo procumbens* could be of medieval origin.

Four monasteries are known from the medieval Trondheim town. No ruins exist above ground of **Elgeseter Augustinian Monastery (c. 1160)** and **The Dominican Convent (c. 1234)**. The last one, located 200 metres NE of Nidaros cathedral, probably had gardens in the same area, where Sandvik (2008) has found plant macrofossils indicating cultivation. Remnants of the church of **Bakke Benedictine nunnery (c. 1150)** still exist in the basement of the present building at Bakke, and ruins of the **Franciscan Convent (c. 1430)** can be seen in the public library.

In the old Norse laws, c. 1000–1300, different forms of gardens are mentioned several times. The law of Frostating includes hazelnuts and leek garden (*laukgarðr*), probably including different species of *Allium*, apple and hop. Bjarkøyretten, Trondheim's city law, regulates cultivation of garden plants in the city, Angelica garden (*kvanngarðr*), cale (vegetable) garden (*kálgarðr*) and leek garden (*laukgarðr*); and in Magnus Lagabøte's law, leek garden, angelica garden, apple garden, turnip garden, pea garden, bean garden and everything that can be cultivated and fenced in are listed (Fægri 1987).

A garden (grasgarð) belonging to the canons in Nidaros is mentioned in 1311, and in another letter dated 1341, a garden near the Clemens church should include hop and be tended by a gardener. Further, he should mow hay (Nøvik 1901).

Some macrofossil finds of possible medieval monastic garden plants from Trondheim include *Cannabis*, *Carum carvi*, *Digitalis purpurea*, *Hyoscyamus niger*, *Iris*, *Linum usitatissimum*, *Malus*, *Papaver*, *Prunus* and *Pyrus* (Sandvik 2006; Östman et al. 2016).

The area just in front of **Selje Benedictine monastery (c. 1100)**, on the small island Selja, is dominated by a conspicuous large plain with traces of cultivation. Pollen analyses revealed cultivation of cereals when the monastery was in operation. At the time, the landscape opened up, and a meadow vegetation rich with herbs was present in addition to the cereal fields. Pollen of *Allium* was present before and during the monastic epoch, but not after. This could originate from cultivation, and both wild onion (*Allium vineale*) and ramsons (*Allium ursinum*) are present at Selja today. Ramsons could possibly be a monastic relict. Pollen of *Plantago major*, *Urtica* and *Artemisia* relates to culture (Hjelle et al. 2009). Possible monastic relict plants found in the ruins include *Aegopodium podagraria*, *Allium ursinum*, *Aquilegia vulgaris*, *Arctium*, *Bellis perennis*, *Urtica dioica* and *Veronica beccabunga*.

It has been suggested that the monastic gardens were placed in a warmer and more protected locality on the island's southern side, called Heimen. Possible monastic relicts here are *Arctium*, *Digitalis purpurea*, *Humulus* and *Verbascum thapsus*. In addition, a particular space in Heimen had once been called Kvanngarden, literally *The Angelica garden* (Høeg 1974). This could likely be a remnant of the monastic gardens at Selja.

Today, the convent church of **The Franciscan Convent (c. 1250)** in Bergen stands as the city's cathedral. In the times following the Reformation, it was said that the Franciscan garden was unequalled in Bergen, although this was based on post-medieval observations. In 1856 it was non-existent (Lange 1847). A group of bird cherries (*Prunus padus*) was planted by the Franciscans, according to local tradition described in 1760 (Hartvedt and Skreien 2013). The plantings are reflected in the present street name, Heggebakken.

Recent archaeological excavations just outside the southern range of the convent indicated possible occurrence of garden soil or an area associated with gardening and agriculture dated approximately the period 1200–1300 (Dunlop et al. 2017). It is not unlikely that the Franciscan garden was placed here, as pollen analyses indicated an area characterized with grass and herbs, including *Artemisia*, *Brassicaceae*, *Urtica*, and possibly *Allium ursinum* (Overland 2016). All could indicate medieval gardening.

No physical trace of the **Dominican Convent (c. 1245)** in Bergen is known today. In approximately 1600, an old beech (*Fagus sylvatica*) was still standing near the convent (Edvardsen 1951). The tree was felled by a storm in 1778. It could have been part of medieval plantings.

**Munkeliv monastery (c. 1110)**, placed on top of Nordnes, was originally Benedictine, but housed Birgittinian nuns in approximately 1420. The monastery possessed extensive meadows in the surrounding area, suitable for agriculture and gardening. A barn near the monastery graveyard is mentioned in the saga of king Sverre (Gundersen 1996). And possibly nearby post-medieval gardens are remnants of the monastery gardens (Lange 1847; Olafsen 1898).

The cult of St. Dorothy—the patroness of gardeners—was practised in Bergen, with an altar to her in the church of the Black Monks (Cormack 2000).

The Agustinian **Jonskloster (c. 1150)** was located east of Munkeliv. The street named Jonsvollgaten is a reminder of the monastery, and the adjacent area Engen (meadow) is probably named after the fields associated with the monastery (Hommedal 2011).

**Nonneseter (Benedictine nunnery c. 1140–1507, St. Anthony order 1507–1528),** with remnants of two structures from the monastery church, is still standing near Bergen railway station, and was developed into a manorial estate from 1528 (Lange 1847). In approximately 1870, several large ash trees dominated the centre of the cloister garth, with two still standing in 1893. A previous nearby garden was eradicated at the time. Both ash trees and garden could possibly have been parts of the monastery gardens (Bendixen 1893; Bruun 2007; Schnitler 1915). Today, nothing remains.

Excavations in 2006 revealed a stone terrace, dated c. 1250, along the southern monastery range, possibly a foundation for a garden. Several layers on top of the terrace have been interpreted as garden soil. Further, in the eastern part, the excavations indicated sence of presence of a monastery garden that later developed into part of the manorial gardens (Reinsnos 2009). Pollen analyses from the area showed an open and grazed meadow vegetation in medieval times, including herbs like *Artemisia vulgaris*, *Centaurea*, *Persicaria maculosa* and *Plantago lanceolata*. In addition, pollen of *Cerealia* and *Humulus/Cannabis* were found, possibly from local cultivation (Halvorsen 2009). The nearby local place name Marken (meadow) reflects the original monasterial fields (Hommedal 2011).

No doubt an extensive gardening activity was present in Bergen in medieval times, and this includes orchards and cultivation of berries, vegetables and herbs (Olafsen 1898; Øye 2015). At least 20–30 different pollen and macrofossil records of possible cultivated utility plants have been recorded, e.g., *Brassica*, *Cannabis*, *Carum carvi*, *Fagopyrum esculentum*, *Humulus*, *Linum usitatissimum*, *Malus sylvestris*, *Malva*, *Mentha*, *Papaver*, *Pisum sativum*, *Prunus cerasus*, *Ribes* and *Vicia faber* (Åsen 2015).

**Lyse Cistercian monastery**, located 20 km south of Bergen, was founded in 1146 as a daughter monastery of Fountains Abbey, England. In 1670, the area including the monastery was privatized and new gardens were established. Today, the land is in private hands and included in the Lyse manor estate. Lyse Monastery also had activities in Opedal in Ullensvang municipality, including a farm, *grangie* (Lidén 2014). The fruit growing,

including apples, plums, pears and cherries in the Hardanger area, may originate from the monks in Opedal according to Olafsen (1900a). Lyse founded Tautra monastery in Trøndelag (see above), and likewise advocated the fruit growing in that area, with apples and sour cherries.

A covered passage led to the infirmary buildings east of the northern range, and a possible infirmary garden could have been placed here, but little is known of the history of the monastery, and gardens are not known. The ruins are located in the middle of an agrarian landscape, where gardening and agriculture could have taken place in medieval times.

Lyse Monastery is well known for its lush growth of masterwort (*Peucedanum ostruthium*), first recorded in 1908. Today, the plant grows fairly close to the actual ruins, together with the Martagon lily (*Lilium martagon*). Masterwort may be a true monastic relict, or it could be a garden escape from the manorial garden dating from the 17th century. Lundquist (2005) has shown that the Martagon lily is a post-medieval introduction to Norway. The masterwort is known from several collections in the Bergen area from 1909 onwards.

**Halsnøy Augustinian monastery (c. 1163)** in Sunnhordland had extensive landholdings in the southern part of Norway, and the estate became a manor after the Reformation. The monastery gardens have most likely been situated outside the buildings. According to Lange (1847), the gardens were in a good condition when he visited the ruins in 1843. He described the monastery buildings based on a drawing and stated that the inner courtyard was planted with ash, of which two were still standing alongside the church wall. A garden—also designated a monastery garden (Lidén 1967)—was marked south of the church on the drawing, and the accompanied text stated that a large kitchen garden was located alongside the beach, south of the ruins (Lange 1847).

Documents dated c. 1750 described an enclosed herb garden outside the main gate of the monastery (Lidén 1967). The herb garden is visible south of the "school" on a painting by Fiigenschough 1656 (Nerhus 1957). The painting also pictures a lot of trees (ash?) around the monastery. South and west of the manor building, three partly walled gardens have been described (Hvinden-Haug and Meyer 2018; Lange 1847; Nøvik 1901). These walls may have originated in medieval times and indicate a continuity from medieval gardening.

Today, Halsnøy monastery with surroundings have a parklike appearance, with extensive lawns and tall trees. Ash is common and was probably once cultivated in a separate garden. A protected ash tree stands in the middle of the ruins. The trunk is hollow, with a circumference over 7 m, and approximately 12 m in height. With a suggested minimum age of approximately 525 years, the old ash is probably a true medieval relict (Moe 2000).

Other possible monastic relict plants recorded at the Halsnøy site include *Aegopodium podagraria*, *Allium ursinum*, *Aquilegia vulgaris*, *Bellis perennis*, *Corylus avellana*, *Crataegus monogyna*, *Malus sylvestris*, *Prunus cerasus*, *Ribes*, *Sambucus nigra* and *Sanguisorba officinalis*.

The present park at **Utstein Augustinian monastery (c. 1265)**, north of Stavanger, is an 18th-century manorial development of the monastery plantings. In 1758, the there was the presence of large ash trees and wild cherries like the ones growing at Tautra (Wagner and Johansen 2019) that could likely be medieval relicts, once growing in an orchard placed east of the monastery buildings.

**Gimsøy Benedictine nunnery (c. 1150)**, in Skien, was a powerful spiritual centre between Tønsberg and Stavanger. In referring to nunnery gardening at Gimsøy, we know nothing. Archaeological excavations in 2007 yielded no traces of either monastery, graveyard or gardens (Meyer and Molaug 2007).

From 1666, the property became part of Gimsøy manor and extensive manorial gardens were established, part of it long referred to as "*The old garden*" (Johnsen 1982; Schnitler 1915). Orchards were important parts of the manorial gardens (Skard 1938) and in 1789, an eyewitness described trees weighed down by apples (Mumssen 1789). This could have been a continuation of possible nunnery gardens. Apple is known from medieval place names in Telemark county, and we may speculate that the knowledge of fruit cultivation could have spread from the nunnery (Olafsen 1900b).

The following list contains possible medieval relict plants growing on Klosterøya today, the island where the Gimsøy nunnery was once located: *Aegopodium podagraria*, *Aethusa cynapium*, *Aquilegia vulgaris*, *Arctium lappa*, *Artemisia vulgaris*, *Bellis perennis*, *Berberis vulgaris*, *Carum carvi*, *Chelidonium majus*, *Fumaria officinalis*, *Hesperis matronalis*, *Lamium album*, *Pastinaca sativa*, *Saponaria officinalis*, *Solanum dulcamara*, *Tanacetum vulgare*, *Urtica dioica*, *Verbascum* and *Viola odorata*.

In the town of Tønsberg, there were two religious houses, a **Franciscan convent (c. 1250)** and **St. Olav's Premonstratensian monastery (c. 1190)**. No ruins are visible above ground of the Franciscan convent. The St. Olav's round church and other ruins after the Premonstratensians are located in the centre of Tønsberg.

Tønsberg was a town of approximately 1500 inhabitants in 1300, with thriving trading activities, and warehouses tightly packed in the harbor. Adjacent open grounds (grasgarð) were exploited for gardening, hay and grazing (Johnsen 1929; Lindh et al. 1984).

In 1277, an agreement between king Magnus Lagabøte and bishop Jon Raude that took place in the Franciscan convent stated, among other things, that tithe should be paid of all fruit (presumably apples, cherries, and may be plums), and of rye, wheat, flax, hemp, turnip and peas. This indicates that these plants were common in cultivation in Norway.

According to a royal letter from 1551, the Franciscans had a garden called "Munke-lykken" (Lange 1856), with cherries and some vegetable beds (Johnsen 1929).

Further, the Premonstratensians had a similar enclosed field associated with their monastery, probably including gardens (Johnsen 1929), and there was room for gardens inside the walled monastery precinct. Pollen samples from the monastery yielded high values of possible cruciferous vegetables or just arable weeds. Pollen of broadbean (*Vicia faba*) suggests cultivation. Other finds include barley and wheat, in addition to *Artemisia*, *Fraxinus excelsior*, *Plantago major*, *Urtica* and *Valeriana* (Hjelle 1988).

Other gardens in Tønsberg (often associated with the church) are mentioned in various texts from the period 1300–1651: hop gardens, grass or hay gardens and flower gardens (blomegard) (Nøvik 1901; Johnsen 1929).

Botanical analyses of pollen and macrofossils supply a long list of utility plants that grew in Tønsberg in the Middle Ages (Eriksson 1990). Some examples of possible monastery relict plants from central parts of Tønsberg include *Allium oleraceum*, *Anchusa officinalis*, *Arctium*, *Artemisia*, *Berberis vulgaris*, *Campanula rapunculoides*, *Carum carvi*, *Chelidonium majus*, *Cichorium intybus*, *Hesperis matronalis*, *Humulus lupulus*, *Hyoscyamus niger*, *Sambucus nigra*, *Solanum dulcamara*, *Solanum nigrum*, *Tanacetum vulgare*, *Verbascum* and *Viola odorata*.

In the Middle Ages, Hamar bishopric was an ecclesiastical centre, including cathedral, the bishop's residence, **the Olavskloster Dominican convent** and hospital (Jordåen 2006). From this area, we have a late medieval description of Norwegian monastic gardens: "*This convent, with its building and location, with orchards, apple—and cherry-gardens, hop-gardens and other glorious facilities, was handsomely and favourable built.*" Further, the bishop's castle had apple and hop gardens, and on a small island in the lake Mjøsa, the bishop cultivated small trees and herbs. The farms located near the convent had all vegetable and herb gardens, orchards with apples and cherries and hop gardens.

All the herbs of Hamar gave a pleasant fragrance, and the pilgrims to Rome and Jerusalem did their best to bring back sweet-smelling herbs. At the time (late medieval), the inhabitants of Hamar loved the rose called the eglantine (*Rosa rubiginosa*), with its pleasant fragrance (Pettersen 2012).

Archaeobotanical samples from a fire, dated 1567, contained *Brassica oleracea*, *Linum usitatissmum*, *Marrubium vulgare* and *Pisum sativum* (Jessen 1956), and all have most likely been cultivated in medieval Hamar.

To sum up, possible monastic garden plants from Hamar may also include *Aquilegia vulgaris*, *Arctium*, *Artemisia absinthium*, *Chelidonium majus*, *Glechoma hederacea*, *Hesperia matronalis*, *Humulus lupulus*, *Hyoscyamus niger*, *Hyssopus officinalis*, *Malus domestica*, *Prunus cerasus*, *Tanacetum vulgare*, *Verbascum* and *Ribes rubrum*, *Ribes uva-crispa*, *Rosa rubiginosa* and *Sambucus nigra*.

The nuns at **Nonneseter Benedictine nunnery (c. 1186)** in Oslo had their own home farm (Inntjore 2000), and gardens could have been part of the farm. Further, the nuns had other properties that point to gardening in approximately the period 1300–1400 (Inntjore 2000).

Following the Lutheran Reformation, parts of **Olavsklosteret Dominican convent (c. 1239)** in Oslo were reused as a school and residence for the bishop (Ekroll 2011). In 1546, the buildings, together with churchyard and garden, were transferred to the latin school (Nøvik 1901). The schoolyard was designated the monks' garden (Schnitler 1915). In approximately 1627, the school moved, the bishop resided in the eastern part of the convent (Ekroll 2011), and he was keen on keeping a herb garden and vegetable garden just like his predecessors (Nøvik 1901).

We sense that these gardens are old and may originate in the Dominican gardens. When the area was excavated in 1927, probably parts of the bishop's garden were found as well as a dam (Fischer 1928). Most likely these structures once constituted the gardens and fish dam of the Dominicans (Bruun 2007).

The citizens of Oslo acquired **the Franciscan convent (founded c. 1285)** in Oslo as a hospital after the Reformation. The buildings burned down in 1567. However, the church stood until 1794, when the present church was erected, partly in connection with the convent church (Inntjore 2000; Nedkvitne and Norseng 2000).

A regular cloister square with Oslo hospital to the north and gardens on the southern side is shown on a 18th-century map (Schnitler 1915; Schia 1991). In 1737, fields, garden and vegetable garden are mentioned connected to the area (Nøvik 1901). This indicates a gardening tradition. The Franciscans in general were invited to grow fruit and vegetables for their own use (Rasmussen 2002). In Oslo, they owned several enclosed fields (Øye 1998); and in 1453, the Franciscans had their own farm in Oslo (Digernes 2010). (Oslo Hospital shut down in 2018.).

From 1970 onwards, several archaeobotanical investigations were carried out in the ecclesiastical centre in Gamlebyen, the old medieval town of Oslo, not precisely at any of the monastery sites, but close by (Griffin 1977, 1979, 1988; Høeg 1977, 1979, 1987), and recently archaeobotanical records from the expansion of the railway through Gamlebyen have been published (Buckland and Wallin 2017; Moltsen 2015, 2016a, 2016b; Sture 2017a, 2017b, 2017c; Wallin 2017).

These records give a general idea of the kind of plants that grew in the old medieval town, both as wild plants or as possible cultivated utility plants. In summary, we have approximately thirty pollen and macrofossil records of possible garden plants, some still growing in the area, e.g., *Allium*, *Arctium*, *Cannabis sativa*, *Chelidonium majus*, *Conium maculatum*, *Daucus carota*, *Foeniculum vulgare*, *Humulus lupulus*, *Hyoscyamus niger*, *Hyssopus officinalis*, *Lamium album*, *Leonurus cardiaca*, *Papaver somniferum*, *Pastinaca sativa*, *Sambucus nigra* and *Verbascum*.

It is interesting to note that a small figure of St. Dorothy was found during the recent excavations (Nordlie et al. 2020). A similar statue was found during the excavations of Viðeyjarklaustur in Iceland (see above). These findings may indicate that gardening was practised.

The ruins of **Hovedøya Cistercian monastery (1147)** are located on an island with calcareous rocks in the harbor of Oslo. The flora in the ruins have been corrupted by the introduction of so called authentic monastic plants in approximately the period 1950–1960. This makes it difficult, if not impossible, to accept new observations of possible relics growing on the island after 1950, and we must rely on older observations and herbarium specimens.

"Just like other Cistercian monasteries Hovedøya monastery had likely its own garden, and presumably it was located west of the ruins", wrote Nicolaysen (1891). We know nothing of this with any certainty. The parent monastery Kirkstead in England had both fishponds and probably gardens (Aston 2007), and we assume the same at a smaller scale

at Hovedøya. The present landscape suggests monastery gardens located west and south of the ruins, with herb garden closest to the buildings.

A small dam is located 150 m SW of the ruins. Pollen analysis by Høeg (2002) indicates that the dam was established c. 1100. Pollen of bog myrtle (*Myrica gale*) was common. Bog myrtle was an important additive to beer and spirits, and also used against insect pests and bad smells. Other pollen finds of *Artemisia*, *Chenopodium*, *Hedera helix*, *Humulus/Cannabis*, *Malva Plantago* and *Urtica* indicate cultivation.

Hovedøya has longbeen well known for its rich flora, with a long period of floristic exploration, resulting in extensive plant lists, with 56 different species found in the ruins proper. The 19th-century observations of possible monastic relict plants from Hovedøya include *Aethusa cynapium*, *Allium oleraceum*, *Aquilegia vulgaris*, *Asparagus officinale*, *Ballota nigra*, *Berberis vulgaris*, *Chelidonium majus*, *Cynoglossum officinale*, *Glechoma hederacea*, *Humulus lupulus*, *Hyoscyamus niger*, *Lithospermum officinale*, *Primula veris*, *Sambucus nigra*, *Valeriana officinalis* and *Verbascum nigrum* (Åsen 2015).

**Værne Knights Hospitallers Monastery (c. 1220)** in Østfold has ruins located inside the Værne 18th-century manorial park. A long post-medieval history of farming and gardening with park and nurseries makes it difficult to conclude anything with certainty about possible medieval garden and relict plants. No archaeobotanical records exist.

Lange (1847) stated that the gardens of Værne still exist in good a condition. Possibly he meant that the manorial garden was a continuation of the monastic garden, part of which was called "the monks' garden", at least until 1915 (Schnitler 1915).

Infirmary gardens are associated with the Knights Hospitallers, the herbs were used in caring for their patients. The ruins of the parent monastery at Antvorskov in Denmark, one of the richest relict plants' monastic ruins in Denmark, contains several typical relict herbs, of which a few also occur at Værne monastery—*Chelidonium majus*, *Campanula rapunculoides*, *Lamium album* and *Viola odorata* are all indications of medieval gardening.

The ruins at **Dragsmark Premonstratentian monastery (Marieskog) (c. 1230)** at Bokenäset in Bohuslän are located close to large fields, and rather protected in a valley enclosed by ridges. The plain could be a location for possible monastic gardens. The dam where the canons raised carps is located southwest of the ruins. Dragsmark Lutheran church and churchyard are situated on the northern side. Interesting plants in or near the ruins that could indicate monastic garden relics include *Aegopodium podagraria*, *Aquilegia vulgaris*, *Bellis perennis*, *Chelidonium majus* (grows everywhere in the ruins), *Cichorium intybus*, *Glechoma hederacea*, *Ribes uva-crispa*, *Urtica dioica* and *Tanacetum vulgare*.

Probably very old bushes (height 6–7 m) of European spindle (*Euonymus europaeus*) grow alongside the stonewall enclosing the churchyard. The spindle was first mentioned in 1838 (Berg 1895), and according to local tradition, planted by the monks and called "*the foreign tree* or *the monks' tree*") (Holmberg and Brusewitz 1867). It was assumed that the churchyard was originally the monastery garden. Further, cultivation of willows in Dragsmark has been associated with the monastery (Elling 1978). However, nothing conclusive can be said of these statements.

In 1423, a farm called Apildatuften ("*Appleground*"), owned by the monastery, is mentioned (Lange 1847), and this could indicate an orchard. According to an eyewitness account c. 1740, absinth (*Artemisia absinthium*) was growing in the ruins (Berg 1895). The absinth is a fairly good indicator of medieval gardening, and a probable relict of the Premonstratensian gardens. The same can be said of catnip (*Nepeta cataria*), growing by the ruins in 1841 (von Düben 1843). In 1847, Lange writes that the gardens of Dragsmark still exist in a good condition. Obviously, he must have seen some kind of gardens that originated in the medieval monastery gardens.

**The Franciscan Convent (c. 1280)** at Marstrand in Bohuslän was located around the present Lutheran church, this church being the original Fransiscan church (Lange 1847; Aasma 1974). However, diverging opinions on the original location of the convent exist. Carlsten fortress (c. 1658–1888/1993) thrones above the village of Marstrand.

During the Reformation, the convent was transformed into an hospital and a school. We know of nothing concrete with respect to the Franciscans possible gardens in Marstrand. The oldest known map of the village, dated 1644, gives no indication of the whereabouts of the cloister nor gardens. In a detailed map dated 1689, a large kålgård (cale garden or vegetable garden) is located west of the church (Aasma 1974), this could have been a possible location for a convent garden. Further, at the hospital, one might expect a herb garden.

Marstrand possessed an important harbor, fortress, hospital and school after the Lutheran Reformation. These circumstances obviously had a strong influence on the introduction of foreign utility plants, and it is quite impossible to definitely say whether a plant is a remnant of Franciscan gardening or rather from post-medieval activities related to the fortress and hospital. Today, the fortress and the streets of Marstrand abound with cultural relict plants, e.g., *Allium scorodoprasum*, *Chelidonium majus*, *Conium maculatum*, *Hyoscyamus niger*, *Lamium album*, *Malva neglecta*, *Papaver somniferum* and *Petasites hybridus*.

The site of **Kastelle Augustinian monastery (c. 1180)** in Bohuslän is located at Klosterkullen, a mound in an agrarian landscape, with trees lining a central area covered mostly by grass and containing the monastic ruins. Klosterkullen seems like a proper place for a herb garden, and some of the common plants growing there today may be characterized as cultural relics from medieval times, e.g., *Aegopodium podagraria*, *Arctium minus*, *Campanula rapunculoides*, *Carum carvi*, *Glechoma hederacea*, *Primula veris*, *Solanum dulcamara*, *Urtica dioica* and *Veronica beccabunga*. At the parent monastery in Æbelholt, Denmark, both infirmary and herb gardens were present (Møller-Christensen 1982), and we would expect that this knowledge was transferred to their brethren at Kastelle monastery.

An inventory of the monastery, dated 1484, mentions a mustard seed grinder (Vigerust 1991). Mustard seeds were either imported or perhaps harvested from *Sinapis* plants in a monastery garden (see Munkaþverárklaustur above).

On the nearby island, Hisingen, the monastery had a cultivated field called Priorløkken (the prior field; Vigerust 1991), and we may visualize possible monastery gardens here. Further, hop is mentioned in the inventory list from 1484 and may have been cultivated in hop gardens.

## 3. Discussion

The monasteries in both Iceland and Norway were part of the wider Roman Catholic world, mostly showing the same physical layout with buildings surrounding a cloister garth. In Europe, monastic gardens were common (Coppack 2006; Harvey 1990; Hennebo 1987; Landsberg 1998; McLean 1989), and no monastery in the later Middle Ages lacked a herb garden (Meyvaert 1986. The cultivation of medicinal and utility plants was important to meet the material needs of the monastic institutions, and obviously this was true also for the Icelandic and Norwegian monasteries.

From Iceland and Norway, six medieval Norse manuscripts dealing with translated classical Greek and Roman traditional medicine are known, and the works of Hippocrates, Galen and Dioscorides eventually found their way to the North. The most important mediator of this knowledge was the Danish physician and canon Henrik Harpestræng (†1244) who has been attributed *Den danske urtebog* (manuscripts from ca 1300). His work was also heavily influenced by The School of Salerno (10th–12th cent.) and was well known and widespread in the Nordic countries (Larsson 2013). During his pilgrimage to Jerusalem in approximately 1150, the abbot, Nikolaus of Munkaþverárklaustur, North Iceland, acknowledged the good doctors of Salerno. The medical and botanical knowledge from southern Europe was mediated to the North by the monasteries, the monks and nuns being among the few who could read and write throughout the Middle Ages, and it is likely that the influence also included the herb gardens. Quite a few northmen were educated at the continental universities, among them Hrafn Sveinbjarnarson (†1213) from Iceland (Schwabe 2010).

According to the Rule of St Benedict, the monasteries should be self-sufficient, and this implied gardening for food and medicine. According to Coppack (2006), only a few

monastic gardens have been noted in excavations in England, and even if no physical garden has yet been found and excavated in either Norway or Iceland, the sum of the compiled data strongly points to the presence of monastery gardens. However, the data does not allow any conclusions with respect to botanical differences between types of monastery institutions, other than that climate and degree of landscape use since medieval times strongly affect the presence of relict plants. Over 80 utility plants are mentioned in the text, possibly cultivated in Icelandic and Norwegian monastery gardens (Table 1). However, extended archaeobotanical studies are highly needed in order to obtain more reliable data, and therefore it is rather impossible to conclude more with respect to existing historical understanding of the northern monastic garden.

## 4. Materials and Methods

All the Icelandic monastery sites were surveyed for landscape and plants (except Keldnaklaustur—unknown during the field work). The first part of the fieldwork took place in July 2009 by Kjell Lundquist, Inger Larsson, Steinunn Kristjánsdóttir and Per Arvid Åsen, the second part in July 2010 by Kjell Lundquist, Inger Larsson and Samson B. Harðarson, and the third part in June to July 2011 by Per Arvid Åsen. It should be noted that the Icelandic monastic sites today are dominated by farming, leaving little space for herbs to grow and survive. Complete plant lists are published by Larsson et al. (2012). The Norwegian monasteries were surveyed by Per Arvid Åsen during several visits between 2003 and 2014 Åsen (2015).

In order to determine possible medieval relict plants, the results were compared with compiled lists of the garden flora of north-western Europe between 800 and 1540 (Harvey 1990), in total over 250 species, and available archaeobotanical data. Further, the extensive list of c. 250 cultural relict plants in Denmark has been consulted (Løjtnant 2017). In addition, a wide range of native species are known as both edible and medicinal plants, but as common native species with a wide distribution including many monastic sites, it has been rather difficult to connect their presence to monastic traditions with certainty.

**Funding:** This research received no external funding.

**Conflicts of Interest:** The author declare no conflict of interest.

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
