# Peer review of "Medieval Monastery Gardens in Iceland and Norway"

_religions, doi:10.3390/rel12050317_

Round 1

Reviewer 1 Report

Review of “Medieval monastery gardens in Iceland and Norway”

The interesting and well-done review paper is assembling archaeological, historical, religious and archaeobotanical evidence concerning the question of medieval monastery gardens in Iceland and Norway. The paper is concise, well written and well organized and presents all different aspect of gardens related to medieval monasteries and the life of the monastic societies. The cited literature is abundant and the bibliography is in general well prepared and arranged. Few relevant citations should be added.

The paper is acceptable after minor revision.

The main problem is already addressed in the introduction as “no physical garden has yet been found and excavated in either Scandinavia or Iceland” (line 9). All information is coming from written sources and analogies and comparisons with results from other contexts, for example excavations of medieval towns which had revealed an abundant archaeobotanical material or from palynological and modern botanical studies where a direct link to a monastery garden is sometimes difficult to establish with certainty.

Nevertheless, it is inspiring to see what we can conclude on medieval monastery gardens in Scandinavia and Iceland even if the main primary source - excavation results – is still missing. The paper merits publication and can draw further focus on these research questions to be partly resolved by future excavations and studies.

One main criticism is that the paper is cruelly lacking a site map and a comparative list of plant names mentioned in the text (please give them in latin, English, Norwegian and Icelandic language. Concerning the figures, at least, a map showing the discussed monastery sites in Norway and on Iceland seems to me necessary, as most readers will not be familiar with these sites and the geography of these countries.

Few minor changes and corrections are suggested and some remarks have to be made:

Lines 35-38

It is not sufficient to cite only an internet source when discussing the famous monastery plan of St. Gall/Sant Gallen in Switzerland. There is a quite abundant literature concerning the important document and at least some titles should be cited which give access to further literature:

Hecht, Konrad. 2005. Der Sankt Galler Klosterplan, 2nd ed. Wiesbaden: VMA-Verlag

Horn, Walter William and Born, Ernest. 1979. The Plan of St. Gall. A Study of the Architecture and Economy of, and Life in a Paradigmatic Carolingian Monastery, 3 vol. California Studies in the History of Art, 19. University of California Press, Berkeley CA.

Jacobsen, Werner. 1992Der Klosterplan von St. Gallen und die karolingische Architektur. Entwicklung und Wandel von Form und Bedeutung im fränkischen Kirchenbau zwischen 751 und 840. Berlin: Deutscher Verlag für Kunstwissenschaft.

Tremp, Ernst. 2014. Der St. Galler Klosterplan: Faksimile, Begleittext, Beischriften und Übersetzung. St. Gallen: Verlag am Klosterhof

 Line 43:

Why citing only violets as example? There are more important symbolic plants for virgin Mary, for ex. lilies and strawberries. The most relevant publication to symbolism of “religious” plants is not cited.

Old, but still important standard books concerning religious plant symbolism:

Behling, Lottlisa. 1957: Die Pflanzen in der mittelalterlichen Tafelmalerei. Weimar 1957, re-edition 1967. Köln, Graz: Böhlau-Verlag.

Behling, Lottlisa. 1964: Die Pflanzenwelt der mittelalterlichen Kathedralen. Köln, Graz: Böhlau-Verlag.

Line 60

…“where the ard plough was the main tool”. This seems to be correct for prehistoric and probably Viking Age times, but in Late medieval times the turning plough was established and allowed a more intense soil preparation for agriculture. Please check and give a time span when the ard was the dominating plough.

Lines 69-72

Please cite also the review article of K.L. Hjelle:

Hjelle, Kari Loe. 2007. Foreign trade and local production ‒ plant remains from medieval times in Norway. In: Medieval Food traditions in Northern Europe. Edited by Sabine Karg. Publications from the National Museum. Studies in Archaeology and History, 12. Copenhagen: National Museum of Denmark, pp. 161-179 and general bibliography pp. 191-215.

Line 79:

Faegri (1987) discusses the possibility that columbine (Aquilegia vulgaris), absinthe (Artemisia absinthium) and and hyssop (Hyssopus officinalis) were introduced to Norway via the monasteries. Nevertheless, There are no archaeobotanical records so far (cited by Hjelle 2007, p. 172).

Line 160:

Please avoid double brackets. Better: (Carum carvi; Larsen et al. 2012).

Line 204 :

Flixweed was not “cultivated in Europe with many archaeobotanical finds”; This sounds very strange. Please reconsider phrasing!

Line 205-206

It is not astonishing that there are no pollen records of Descurainia sophia, flixweed, as most pollen from plants of the mustard family (Brassicacae) cannot be determined to species level.

Line 211-212

I do not like the creation of the term “monastic utility plants”; I would simply speak of useful plants estimated to be plantes/cultivated in monastery gardens. Nevertheless, most species are common wild plants and nothing is really proving their cultivation or introduction if they are generally common on Iceland. If this is not the case, please argue in detail.

Line 251:

Are the stinging nettles Urtica dioica and Urtica urens absent from natural and semi-natural vegetation in Iceland or which argument speaks in favour of an introduction as medicinal herbs? Please explain. Why are they not simply growing in the semi-natural ruderal vegetation around the monastery?

Line 252:

Import of carb apple as a food seems to me quite ridiculous – gustative value is low, carb apple must be cooked or boiled for consumption. Determination seems to be in question – so a quite doubtful evidence…

Line 390-391

Please avoid the term “monastic plants”. They are simply useful plants, maybe cultivated in relation with a monastery, but nothing else qualifies them to be “monastic”. Please delete “monastic” as these plants occur also general in urban and suburban contexts.

Line 437-439:

The latin plant names must be given in italics.

Line 463:

Bittersweet nightshade Solanum dulcamara is a typical wetland plant and absolutely not suitable to be cultivated in a monastery garden.

Line 493:

Solanum nigrum, black nightshade, is a typical weed of cereals and other cultivated crops, which occurs also in ruderal vegetation. It is always common in archaeobotanical assemblages from various contexts. Why this should be a typical medieval relic plant? Arctium and Artemisia are also very common in ruderal vegetation during the medieval and early modern period. I guess, it is necessary to argue in more detail.

Line 509:

Marrubium vulgare is not producing seeds, but achenes/nutlets.

Line 511-515:

The latin plant names must be given in italics.

Line 531:

“Dominicans” instead of “Domincans”

Line 656:

Please avoid double brackets. Better: (the prior field; Vigerust 1991)

Line 487:

The “material and method” heading after the discussion makes to me no sense. Better place this before the detailed site descriptions.

The paper is acceptable for the journal “Religions” (special issu on medieval monasteries) after minor revision and needs only few improvements. It is a very appreciated survey and well-done review of the available evidence of medieval monasteries and possible monastery gardens in Iceland and Norway, demonstrating a deep knowledge of the research subject and all types of interdisciplinary approach.

Metz, 14.04.2021

Dr. Julian Wiethold

Laboratoire archéobotanique

Institut national de recherches archéologiques préventives (Inrap)

[email protected]

Tel. 0033/387162251

Reviewer 2 Report

Comments are attached.

Author Response

Please see attachment!

I have included my responses in the article as best as I could.

Reviewer 3 Report

Peer review

Medieval monastery gardens in Iceland and Norway

Following an introduction to medicinal plants, the author systematically reviews all monastic plots in Iceland and most of the Norwegian plots. In this connection, all known relic plants that have been found in the various places are listed. The study concludes with a discussion pointing out the connection between the monasteries in Iceland and Norway with the rest of Europe as well as an account of the materials and methods of the study.

The article contains references to a very comprehensive collection of literature.

The article meets the academic requirements for that type of work.

Author Response

OK, thanks.

Round 2

Reviewer 2 Report

The clarifications are helpful.  The author might consider setting out the key research questions and the scope of the article at the very beginning of the paper, before the description of the history of the monastic garden.  This might stress the originality and importance of the approach for elucidating the character of the northern monastic garden, and stress that 80 species of plants have been identified that may be linked to monastic origins.